

# Soil organic carbon pools and carbon management index under different land use systems in North western Himalayas

Yasir Hanif Mir[1], Mumtaz Ahmad Ganie[2], Tajamul Islam Shah[3], Aziz Mujtaba Aezum[3], Shabir Ahmed Bangroo[3], Shakeel Ahmad Mir[3], Shahnawaz Rasool Dar[4], Syed Sheeraz Mahdi[5], Zahoor Ahmad Baba[6], Aanisa Manzoor Shah[1], Uzma Majeed[7], Tatiana Minkina[8], Vishnu D. Rajput[8] and Aijaz Ahmad Dar[9]

[1] Division of Soil Science and Agricultural Chemistry, Faculty of Agriculture, SKUAST-Kashmir, Wadura, Jammu and Kashmir, India
[2] KVK Shopian, SKUAST-Kashmir, Shopian, Jammu and Kashmir, India
[3] Division of Soil Science, SKUAST-Kashmir, Srinagar, Jammu and Kashmir, India
[4] Research Center for Residue & Quality Analysis, Sheri Kashmir University of Agricultural Sciences & Technologies, Kashmir, Srinagar, Jammu and Kashmir, India
[5] Division of Agronomy, Faculty of Agriculture, SKUAST-Kashmir, Wadura, Jammu and Kashmir, India
[6] Division of Basic Sciences and Humanities, Faculty of Agriculture, SKUAST-Kashmir, Sopore, Jammu and Kashmir, India
[7] Division of Agricultural Statistics, Faculty of Horticulture, SKUAST Kashmir, Srinagar, Jammu and Kashmir, India
[8] Academy of Biology and Biotechnology, Southern Federal University, Oblast, Russia
[9] Directorate of Planning, SKUAST-Kashmir, Srinagar, Jammu and Kashmir, India

Corresponding authors
Tajamul Islam Shah,
tajamulislam@skuastkashmir.ac.in,
3tajamul@gmail.com
Shahnawaz Rasool Dar,
srdarskuastk@gmail.com

## ABSTRACT

Current study was conducted to evaluate the effect of important land uses and soil depth on soil organic carbon pools viz. total organic carbon, Walkley and black carbon, labile organic carbon, particulate organic carbon, microbial biomass carbon and carbon management index (CMI) in the north Western Himalayas, India. Soil samples from five different land uses viz. forest, pasture, apple, saffron and paddy-oilseed were collected up to a depth of 1 m (0–30, 30–60, 60–90 cm). The results revealed that regardless of soil depth, all the carbon pools differed significantly ($p < 0.05$) among studied land use systems with maximum values observed under forest soils and lowest under paddy-oilseed soils. Further, upon evaluating the impact of soil depth, a significant ($p < 0.05$) decline and variation in all the carbon pools was observed with maximum values recorded in surface (0–30 cm) soils and least in sub-surface (60–90 cm) layers. CMI was higher in forest soils and lowest in paddy-oilseed. From regression analysis, a positive significant association (high R-squared values) between CMI and soil organic carbon pools was also observed at all three depths. Therefore, land use changes and soil depth had a significant impact on soil organic carbon pools and eventually on CMI, which is used as deterioration indicator or soil carbon rehabilitation that influences the universal goal of sustainability in the long run.

# INTRODUCTION

SOC is a substantial fertility attribute that affects various soil properties, thus influencing the quality of soil and ecological functions (*Benbi et al., 2015*; *Yadav et al., 2018*). It tends to sequester carbon and performs a significant function in the global carbon cycle (*Lal, 2018*) and climate change (*Zhang et al., 2017*). However, anthropogenic disturbances have been a significant reason for soil loss and degeneration of carbon stocks, which is a grievous concern to ecosystem sustainability (*Yadav et al., 2018*). Soils have undergone unabated degradation at a formidable rate owing to wind and water erosion, desertification, and salinization resulting from abuse and inapt agricultural practices. Therefore, it becomes critical to protect them from further deterioration as there is a corresponding depletion in soil quality to produce nutritious crops. Soil organic matter (SOM) is fundamental for agricultural sustainability and its compositional alterations could eventuate in both total and active carbon pools (*Blair, Lefroy & Lisle, 1995*) where labile fraction includes microbial biomass carbon (MBC), particulate organic matter (POM), easily extractable carbon, readily mineralizable carbon and carbohydrates (*Haynes, 2005*). Among the fractions, labile organic carbon (LOC) is quite susceptible to the alterations in vegetation and microenvironment of soil and management practices (*Sahoo et al., 2019*; *Cheng, Chen & Luo, 2008*; *Sainepo, Gachene & Karuma, 2018*). Soil carbon is substantially impacted by climate, and land use changes (*Lal, 2018*), and the transition from natural to the managed system causes SOC pool depletion (*Yigini & Panagos, 2016*; *Somasundaram et al., 2018*; *Meena et al., 2018*). In the Kashmir Himalayas, a significant impact of cropping sequences on carbon sequestration has been observed, resulting in the alteration of soil physical, chemical, and biological properties (*Kalambukattu et al., 2013*). In addition, conservation tillage and agro-forestry system have been found to have a greater ability for SOC storage among the different management practices in agricultural soils (*Babu et al., 2020*; *Luo, Wang & Sun, 2010*; *Lal, 2002*; *Nath & Lal, 2017*). Soil stores four times more carbon than the biosphere up to 2 m (*Lal, 2001*) and fewer alterations in SOC can aid in significant variation in atmospheric carbon dioxide stock (*Babu et al., 2020*). Thus sustaining SOM is of paramount significance for conserving the quality of soil and ecological balance.

The physical, chemical, and biological properties of soil and the self-organization capacity are directly impacted by organic carbon (OC) pools and carbon lability (*Addiscott, 1995*; *Blair & Crocker, 2000*). As a result, their integration into CMI, could serve as a vital metric for assessing the potential of management systems for improving soil quality (*Blair, Lefroy & Lisle, 1995*; *Blair et al., 2006a*; *Blair et al., 2006b*; *Diekow et al., 2005*). Greenhouse effect could also be reduced by sequestering carbon through suitable land-use systems (*Sofi et al., 2016*; *Han, Li & Horwath, 2013*). Forest land use has a maximum potential to sequester soil carbon compared to other land uses (*Kooch et al., 2012*). However, the prominent land uses of Kashmir valley are under constant stress caused by land conversions owing to cultivation and infrastructural activities (*Fayaz et al., 2020*), thus reducing carbon
sequestration and threatening sustainability. Moreover, most of the studies are concerned with fertility evaluation and crop yields, water, and fertilizer management, however, the impact of LUSs and soil depth on SOC fractions and CMI in the Kashmir Himalayas has not been well studied. Pertaining to the significance of related work, proper comprehension and consideration of research regarding carbon pools and carbon sequestration are needed. Hence, this study was carried out to (i) determine the influence of different LUSs and soil depth on SOC pools, (ii) to evaluate the CMI among LUSs at different depths, and (iii) to identify the relationships between various C fractions and CMI. The outcome is expected to be substantial in order to develop strategies for better soil carbon management and carbon rehabilitation to serve the purpose of global sustainability.

# MATERIALS AND METHODS

## Study area

A study was carried out across Kashmir Valley in India with an altitude range from 1587–2640 m. Kashmir Valley lies between the coordinates of 33°20′ to 34°54′N and 73°55′ to 75°35′E occupying a total area of 1.5948 million hectares, which includes the net sown area (3.31 lakh hectares), forest (5.24 thousand hectares), permanent pastures (34.92 thousand hectares) and other grazing lands. The soils in the study area are of different groups including hapludalfs, hapludolls, ochraualfs, eutrochrepts, croboralfs, argiudolls, ustifluvents, udifluvents, fluviatile and lacustrine. Five different land uses were selected based on the land use system being followed over the last 30–40 years viz., forest, pasture, apple, saffron, and paddy-oilseed. A purposive random method of sampling was used for the collection of geo-referenced soil samples from preferred LUSs prevailing in the study area. In the forest, soil samples were taken in between the tree rows, in pasture lands, soil samples were taken in the moderately grazed area. However, in apple, soil samples were taken just outside the drip line and in saffron, samples were taken inside corm beds while in the paddy-oilseed cropping system samples were taken inside crop fields. All the samples were collected in the summer season at 0–30, 30–60, and 60–90 cm soil depth. Each soil sample was a composite of four sub-samples, and three soil samples from different locations were collected, for each land use system.

## Physico-chemical properties

Soil samples collected from different locations were air dried, ground and passed through a two mm sieve to analyze different parameters. Particle size distribution was assessed by the hydrometer method in line with *Bouyoucos (1962)*.

Soil reaction (pH) was assessed using a glass electrode pH meter as described by *Jackson (1973)*.

Bulk density (BD) was evaluated by employing the protocols described by *Blake & Hartge (1986)*. Soil cores were taken from each location and from each land use system. The weight of soil + core sampler ($Wt_{sc}$) was recorded and the volume of core sampler (V) was determined, and substituted in the equation:

$$BD = \frac{Wt_{sc}}{V}$$
Available nitrogen (N) was assessed by employing the respective standard method described in *Subbiah & Asija (1956)*, where 5 g soil sample (<2 mm) was treated with 20 ml of 0.32% $KMnO_4$, 25 ml of 2.5% sodium hydroxide solution and 10 ml of distilled water followed by distillation. To determine the nitrogen content of the sample, a distillate was collected in a boric acid mixture, and absorbed ammonia was titrated with 0.2 N $H_2SO_4$.

## Soil organic carbon pools

Total organic carbon represents the carbon stocks in the soil which was determined by the wet oxidation as previously described in *Snyder & Trofymow (1984)*. The calculation is as follows:

Mg C trapped = (mean meq $H^+$ used in titrating blanks − meq $H^+$ used in sample) × 12 mg C/2 meq +

When 1.0 N HCL is used in titrating, this formula reduces to:

Mg C trapped = (ml HCL Blank − ml HCL sample) × 6

Walkley Black carbon (WBC) represents readily oxidizable OC which was evaluated by oxidizing OM using chromic acid, followed by back titration of unconsumed potassium dichromate with ferrous sulfate (*Walkley & Black, 1934*).

For labile carbon determination, the soil was oxidized with 333 mM $KMnO_4$ followed by shaking and centrifugation, and the concentration of $KMnO_4$ was quantified at 565 nm wavelength using a spectrophotometer (*Blair et al., 1997*).

Particulate organic carbon was determined by shaking a suspension (soil + 0.5% sodium hexametaphosphate) for 24 h followed by sieving through 0.053 mm and the retained material was subjected to the wet oxidation method (*Camberdella & Elliott, 1992*).

MBC was determined using the fumigation extraction method as previously described in *Jenkinson & Powlson (1976)*, where one set was fumigated with fresh ethanol-free chloroform while another set was un-fumigated and extraction with 0.5 M $K_2SO_4$ followed by oxidation with $K_2Cr_2O_7$ and titration with ferrous ammonium sulfate.

## Carbon management index

CMI is an evaluation methodology that indicates how a certain land type alters the quality of soil in contrast to a reference land use. The CMI was computed from the following relationships:

(i) Carbon pool index (CPI) $= \dfrac{\text{Sample total C(mg/g)}}{\text{Reference total C(mg/g)}}$

(ii) Carbon lability index was calculated as follows:

(a) Lability of Carbon $= \dfrac{\text{C in fraction oxidized by KMnO4}}{\text{C remaining un-oxidized by KMnO4}}$

The lability of carbon was estimated from the difference between labile carbon content and non-labile carbon content (difference between TOC and LOC) of samples.

(b) Lability index (LI) $= \dfrac{\text{Lability of C in sample soil}}{\text{Lability of C in reference soil}}$

(iii) Then, CMI was calculated as follows:

$$CMI = CPI \times LI \times 100$$

CMI was computed for all the land uses; taking native forest soil as a reference with a known CMI of 100.

## Statistical analysis

A two-way ANOVA was employed to evaluate the influence of land use and soil depth on SOC pools. The statistical difference was determined at $P < 0.05$. A correlation coefficient was performed to evaluate the association among physico-chemical characteristics and SOC pools. A simple linear regression was employed to comprehend the relation between CMI and SOC fractions.

## RESULTS AND DISCUSSION

### Physico-chemical properties of soils under different land use systems

The mean sand percentage at the surface (0–30 cm) and sub-surface (30–60 and 60–90 cm) soils varied from 19.33–54.20, 17.10–50.30, and 15.46–46.37%, respectively, with the minimum content in paddy-oilseed soils and maximum in forest soils (Fig. 1). The silt percentage at the surface (0–30 cm) and sub-surfaces (30-60 and 60–90 cm) varied from 34.20–50.53, 35.43–51.13, and 36.90–52.76% with the least in paddy-oilseed soils and higher in apple (Fig. 1). The clay percentage varied between 11.60–33.70, 14.26–35.00, and 16.73–38.10% with the maximum in paddy-oilseed soils and minimum in forest soils. The studied land uses were categorized as sandy loam, loam, silt loam, clay loam, and silty clay loam. This is aligned with the reports of *Maqbool, Rasool & Ramzan (2017)* and *Mahapatra et al. (2000)*. The sand percentage exhibited a declining pattern along the soil depth while silt and clay percentage depicted an increasing pattern along the soil depth, which might be ascribed to illuviation and translocation of clay from upper soil layers (*Najar et al., 2009*; *Mohamed et al., 2017*). Forest soils exhibited coarse texture with more percentage of sand and a low percentage of clay owing to steep slopes, reduced soil development, higher altitudes, less infiltration, and vice versa in cultivated soils (*Abad, Khosravi & Alamdarlou, 2014*; *Kiflu & Beyene, 2013*; *Maqbool, Rasool & Ramzan, 2017*). The N levels in surface (0–30 cm) and sub-surfaces (30-60 and 60–90 cm) varied from 279.30–417.52, 237.04–313.85, and 167.67–235.40 kg ha$^{-1}$, respectively, with maximum values in forest soils and minimum in cultivated land uses (Fig. 2). A decline was observed in N levels along the soil depth with a significant ($p < 0.05$) difference across the depths (Table 1, Fig. 1). These findings are in line with the previous reports of scientists (*Pal, Panwar & Bhardwaj, 2013*; *Maqbool, Rasool & Ramzan, 2017*). The mean values of BD in surface (0–30 cm) and sub-surface (30-60 and 60–90 cm) soils varied from 1.18–1.38, 1.25–1.42, and 1.31–1.51 Mgm$^{-3}$, respectively, with the minimum values observed under forest soils and maximum under paddy-oilseed soils (Fig. 2). Further, while evaluating the impact of soil depth, an increment was observed and all the examined depths differed significantly ($p < 0.05$) (Table 1, Fig. 2). This could be attributed to the absence of anthropogenic disturbances,

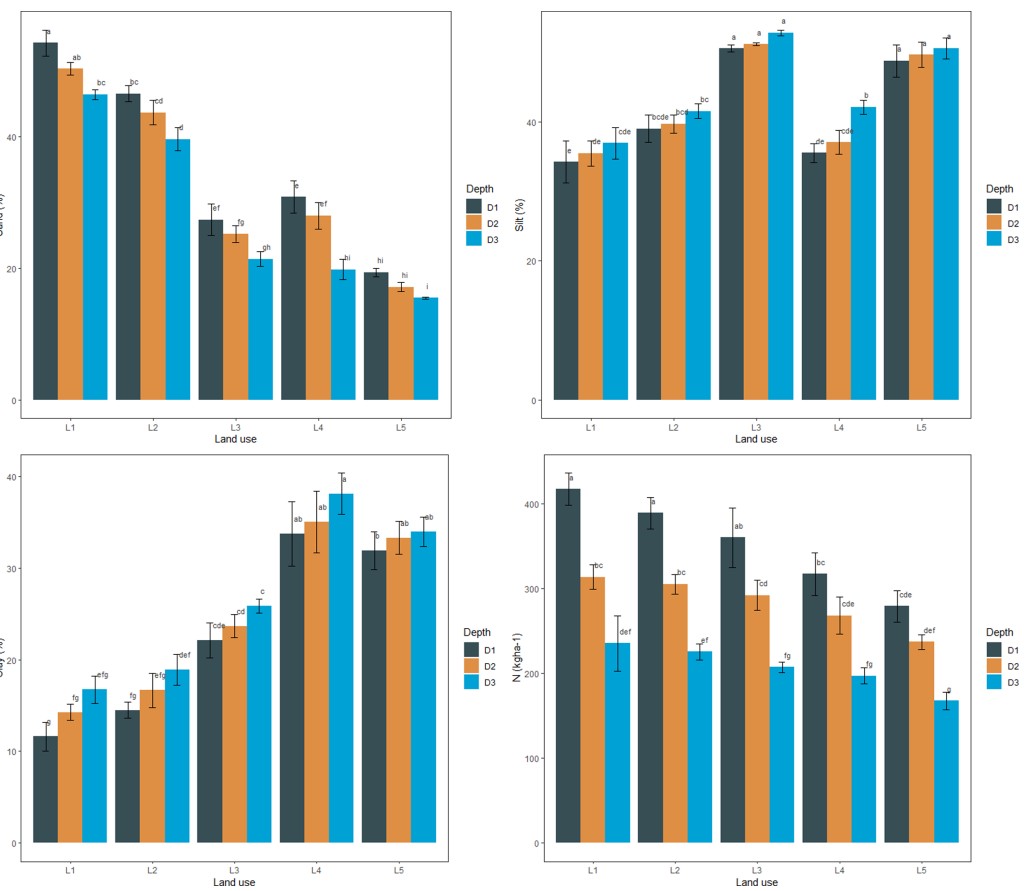

**Figure 1** Particle size distribution and nitrogen content under different LUSs and soil depth.

and higher organic matter content in forest soils and vice versa in cultivated soils. These are further consistent with the previous reports (*Sofi, Rattan & Datta, 2012*; *Chemeda, Kibret & Fite, 2017*). The mean values of soil pH in the surface (0–30 cm) and sub-surface (30-60 and 60–90 cm) layer varied from 6.13–7.17, 6.37–7.30, and 7.33–7.50, respectively, with minimum values observed in forest soil and maximum in paddy-oilseed soils (Fig. 2). Moreover, an increment was observed along the soil depth, and all the examined soil depths differed significantly ($p < 0.05$) (Table 1, Fig. 2). In general, soils were slightly acidic to slightly alkaline in reaction. This might be ascribed to the absorption of bases by tree biomass, the acidic character of litter following its degradation, the accumulation of organic matter in forest soils, and rapid oxidation owing to tillage practices in cultivated soils. Our results agree with the prior reports (*Muche, Addis & Molla, 2015*; *Kiflu & Beyene, 2013*).

## Distribution of SOC pools across studied LUSs

Statistical analysis adopting two factor completely randomized design with interaction was employed for the data obtained from investigated LUSs at different depths (Tables 2–6). Table 2 depicts a two-way ANOVA, reflecting the LUS and soil depth influence on

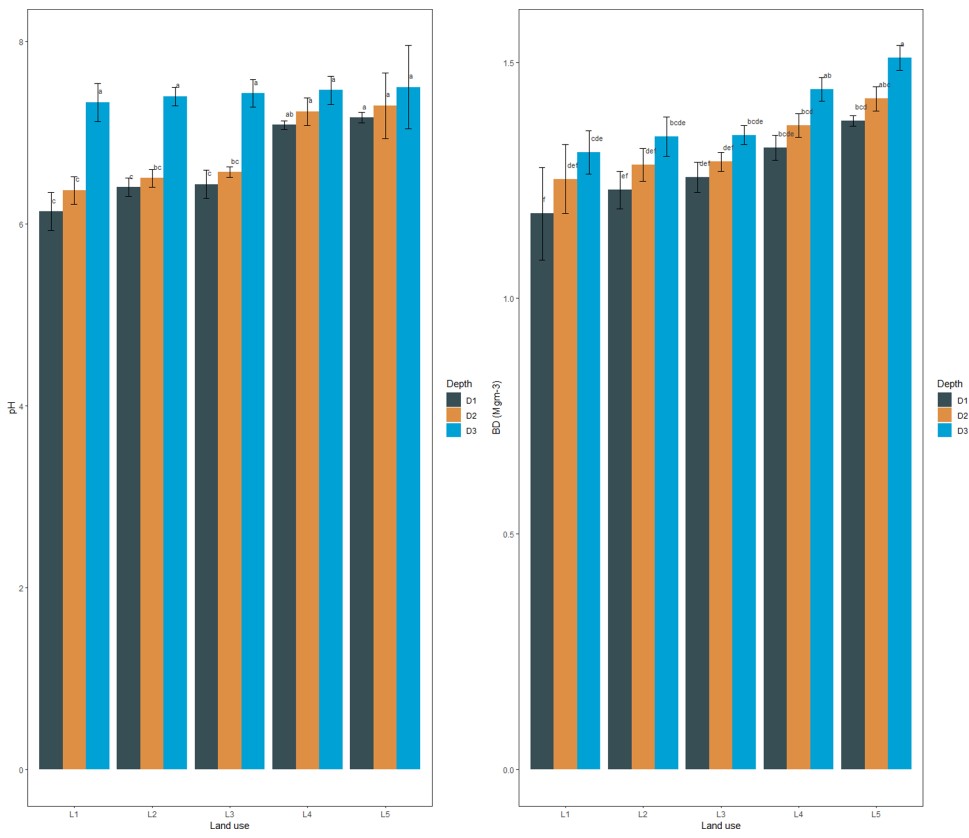

**Figure 2 Soil reaction (pH) and bulk density under different LUSs and soil depth.**

**Table 1 Two-way ANOVA for the effect of land use system (LUS) and soil depth on sand, silt, clay, BD, pH, and N.**

| Treatment | df | Sand | Silt | Clay | BD | pH | N |
|---|---|---|---|---|---|---|---|
| | | $p$ | $p$ | $p$ | $p$ | $p$ | $p$ |
| LUS | 4 | <0.001 | <0.001 | <0.001 | <0.001 | <0.001 | <0.001 |
| SD | 2 | <0.001 | <0.001 | <0.001 | <0.001 | <0.001 | <0.001 |
| LUS × SD | 8 | 0.02 | 0.33 | 0.94 | 0.99 | <0.001 | 0.06 |

TOC. The mean values of TOC under surface (0–30 cm) soils varied from 36.12–64.32 Mg ha$^{-1}$ and in sub-surface (30-60 and 60–90 cm) layers from 27.48–55.66 Mg ha$^{-1}$ and 18.90–44.54 Mg ha$^{-1}$, respectively across studied LUSs. The amount of TOC differed significantly ($p < 0.05$) among studied LUSs with maximum content under forest soils and minimum under paddy-oilseed soils. Furthermore, a significant decline has been observed in TOC content along the soil depth ($p < 0.05$). The increased amounts of TOC in the forest soils might be associated with a significant annual inclusion of organic materials as plant litter that persists in the soil because of no intrusion and seasonal tillage (*Smith, 2007*; *Baker et al., 2007*). The rate of decomposition is impeded by the lower temperature conditions in high altitudes which accounts for increased carbon values. The smaller level of TOC in croplands could be because of the negligible surface cover, removal of biomass

**Table 2  Effect of LUSs on TOC (Mg ha⁻¹) at different depths.**

| Depth/Land use | Depth-1 | Depth-2 | Depth-3 | Factor mean |
|---|---|---|---|---|
| Forest | 64.32 ± 0.63 | 55.66 ± 0.81 | 44.54 ± 0.93 | 54.84[a] |
| Pasture | 59.33 ± 0.34 | 52.58 ± 0.52 | 42.14 ± 0.57 | 51.35[b] |
| Apple | 48.24 ± 0.21 | 39.97 ± 0.10 | 31.00 ± 0.10 | 39.73[c] |
| Saffron | 41.28 ± 0.36 | 30.62 ± 0.33 | 21.73 ± 0.48 | 31.21[d] |
| Paddy-Oil seed | 36.12 ± 0.22 | 27.48 ± 0.28 | 18.90 ± 0.16 | 27.50[e] |
| Factor Mean | 49.85[a] | 41.26[b] | 31.66[c] | |
| C.D ($p < 0.05$) | Land Use = 0.79 | Depth = 0.61 | Land use × Depth = 1.37 | |

**Notes.**

Depth-1 = 0–30 cm  Depth-2 = 30–60 cm  Depth-3 = 60–90 cm
Mean values possessing different letters are significantly different at probability level of $\alpha$ 0.05.

in harvested products, elimination of crop residue, and increased tillage which enhances the loss of carbon (*Smith, 2007*). Moreover, increased TOC levels in the upper layer might be attributed to increased quantities of litter inputs in the surface layer. These findings conform with *Haynes (2005)*, *Smith (2007)*, *Baker et al. (2007)*, *Kalambukattu et al. (2013)*, *Sofi et al. (2016)*, *Giannetta et al. (2018)* and *Giannetta et al. (2019)*.

Table 3 depicts the mean values of WBC under surface (0–30 cm) soils varied from 11.70–22.00 g kg⁻¹ and in sub-surface (30-60 and 60–90 cm) layers from 11.46–18.86 and 10.36–16.36 g kg⁻¹, respectively across studied land use systems. The amount of WBC differed significantly ($p < 0.05$) among studied LUSs with maximum content under forest soils and minimum under paddy-oilseed soils, while saffron and paddy-oilseed were at par. Further, a significant ($p < 0.05$) reduction in WBC was recorded along the soil depth. Higher organic carbon in surface soils of forests has also been recorded by *Miller & Grardiner (2001)*, *Kaleem & Ghulam (2005)*, *Yimer, Ledin & Abdulakdir (2007)*, *Jamala & Oke (2013)* and *Yihenew, Fentanesh & Solomon (2015)*. The low organic carbon in cultivated lands might be due to the rapid mineralization and loss of carbon from soil (*Chauhan, Pande & Thakur, 2014*), long-term cultivation under submerged conditions resulting in stable aggregate breakdown and SOM degradation (*Yang, Yang & Ouyang, 2005*), eventually deteriorates soil quality. The reports are further supported by the results of *Liding et al. (2011)*, *Mojiri, Aziz & Ramaji (2012)*, *Wiesmeier et al. (2012)*, *Poeplau & Don (2013)*, *Chemeda, Kibret & Fite (2017)* and *Kaur & Bhat (2017)*.

The mean values of LOC under surface (0–30 cm) soils varied from 2.33–7.20 g kg⁻¹ and in sub-surface (30-60 and 60–90 cm) layers from 2.09–4.74 and 1.86–3.28 g kg⁻¹, respectively across studied land use systems (Table 4). The amount of LOC differed significantly ($p < 0.05$) among studied LUSs with maximum content under forest soils and minimum under paddy-oilseed soils, following a trend; forest >pasture >apple >saffron >paddy-oilseed. Furthermore, while evaluating the effect of soil depth, a significant decline in LOC content was observed and all depths differed significantly ( $p < 0.05$). The continuous annual inclusion of rapidly decomposable plant litter and increased MBC levels justifies the elevated amount of LOC in forest soils. Whereas the lower values of labile carbon among studied land uses can be attributed to the destabilization of aggregates and accelerated oxidation of SOM in plow and harrow-based traditional cultivation systems

**Table 3  Effect of LUSs on WBC (g kg⁻¹) at different depths.**

| Depth/Land Use | Depth-1 | Depth-2 | Depth-3 | Factor Mean |
|---|---|---|---|---|
| Forest | 22.00 ± 0.11 | 18.86 ± 0.52 | 16.36 ± 0.50 | 19.07[a] |
| Pasture | 16.90 ± 0.52 | 14.73 ± 0.44 | 13.33 ± 0.53 | 14.99[b] |
| Apple | 16.86 ± 0.06 | 13.96 ± 0.87 | 11.73 ± 0.18 | 14.18[c] |
| Saffron | 12.36 ± 0.06 | 11.50 ± 0.30 | 10.80 ± 0.36 | 11.56[d] |
| Paddy-Oilseed | 11.70 ± 0.23 | 11.46 ± 0.32 | 10.36 ± 0.24 | 11.17[d] |
| Factor Mean | 15.96[a] | 14.10[b] | 12.52[c] | |
| C.D ($p < 0.05$) | Land Use = 0.69 | Depth = 0.53 | Land Use × Depth = 1.19 | |

Notes.
Depth-1 = 0–30 cm Depth-2 = 30–60 cm Depth-3 = 60–90 cm
Mean values possessing different letters are significantly different at probability level of $\alpha$ 0.05.

**Table 4  Effect of LUSs on LOC (g kg⁻¹) at different depths.**

| Depth/Land Use | Depth-1 | Depth-2 | Depth-3 | Factor Mean |
|---|---|---|---|---|
| Forest | 7.20 ± 0.38 | 4.74 ± 0.03 | 3.28 ± 0.02 | 5.07[a] |
| Pasture | 6.82 ± 0.32 | 4.06 ± 0.18 | 3.01 ± 0.04 | 4.62[b] |
| Apple | 4.62 ± 0.24 | 3.87 ± 0.04 | 2.12 ± 0.04 | 3.53[c] |
| Saffron | 3.66 ± 0.32 | 3.16 ± 0.16 | 2.05 ± 0.05 | 2.95[d] |
| Paddy-Oilseed | 2.33 ± 0.07 | 2.09 ± 0.02 | 1.86 ± 0.06 | 2.09[e] |
| Factor Mean | 4.92[a] | 3.58[b] | 2.46[c] | |
| C.D ($p < 0.05$) | Land Use = 0.30 | Depth = 0.23 | Land Use × Depth = 0.53 | |

Notes.
Depth-1 = 0–30 cm Depth-2 = 30–60 cm Depth-3 = 60–90 cm
Mean values possessing different letters are significantly different at probability level of $\alpha$ 0.05.

**Table 5  Effect of LUSs on POC (mg kg⁻¹) at different depths.**

| Depth/Land Use | Depth-1 | Depth-2 | Depth-3 | Factor Mean |
|---|---|---|---|---|
| Forest | 1380.24 ± 60.01 | 1122.11 ± 15.30 | 971.11 ± 6.74 | 1157.82[a] |
| Pasture | 1290.34 ± 35.38 | 1091.36 ± 5.76 | 894.14 ± 6.48 | 1091.95[b] |
| Apple | 910.52 ± 9.28 | 788.47 ± 3.68 | 693.13 ± 3.18 | 797.37[c] |
| Saffron | 870.44 ± 14.84 | 673.65 ± 7.87 | 492.42 ± 1.38 | 678.84[d] |
| Paddy-Oilseed | 756.31 ± 19.13 | 569.12 ± 1.41 | 390.74 ± 1.37 | 572.06[e] |
| Factor Mean | 1041.57[a] | 848.94[b] | 688.30[c] | |
| C.D ($p < 0.05$) | Land use = 33.44 | Depth = 25.90 | Land Use × Depth = 57.92 | |

Notes.
Depth-1 = 0–30 cm Depth-2 = 30–60 cm Depth-3 = 60–90 cm.
Mean values possessing different letters are significantly different at probability level of $\alpha$ 0.05.

(*Bayer et al., 2006*). These observations are concordant with those of *Garten Jr, Post & Hanson (1999)*, *McLauchlan & Hobbie (2004)*, *Cheng, Chen & Luo (2008)*, and *Sofi et al. (2016)*.

The mean values of POC under surface (0–30 cm) soils ranged from 756.31–1,380.24 mg kg⁻¹ and in sub-surface (30–60 and 60–90 cm) layers from 569.12–1,122.11 mg kg⁻¹ and 390.74–971.11 mg kg⁻¹, respectively across studied LUSs (Table 5). The amount of POC differed significantly ($p < 0.05$) among studied LUSs with maximum content under forest

**Table 6   Effect of LUSs on MBC (mg kg⁻¹) at different depths.**

| Depth/Land Use | Depth-1 | Depth-2 | Depth-3 | Factor Mean |
|---|---|---|---|---|
| Forest | 1458.12 ± 33.51 | 1232.53 ± 6.08 | 1004.14 ± 2.76 | 1232.59[a] |
| Pasture | 1370.32 ± 108.48 | 1189.52 ± 115.43 | 993.21 ± 102.83 | 1184.35[b] |
| Apple | 1105.16 ± 96.40 | 933.38 ± 102.01 | 771.37 ± 90.86 | 936.64[c] |
| Saffron | 968.26 ± 139.23 | 703.61 ± 130.18 | 515.30 ± 124.80 | 729.05[d] |
| Paddy-Oilseed | 830.21 ± 141.18 | 546.38 ± 136.36 | 365.50 ± 128.10 | 581.03[e] |
| Factor Mean | 1146.42[a] | 921.28[b] | 729.90[c] | |
| C.D ($p < 0.05$) | Land Use = 20.23 | Depth = 15.67 | Land Use × Depth = 35.05 | |

**Notes.**

Depth-1 = 0–30 cm Depth-2 = 30–60 cm Depth-3 = 60–90 cm

Mean values possessing different letters are significantly different at probability level of $\alpha$ 0.05.

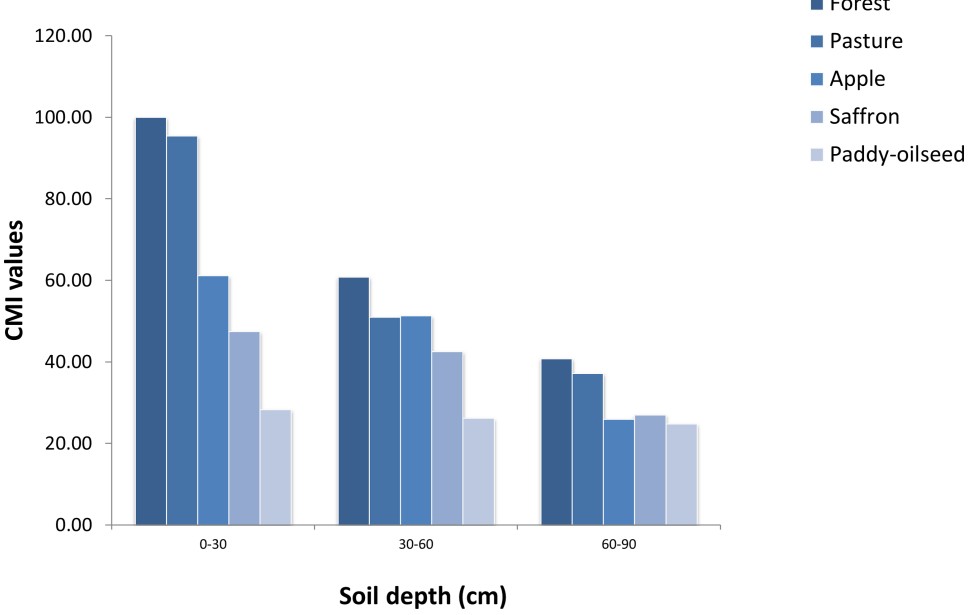

**Figure 3   CMI values of different LUSs at varying depths (Reference land use: Forest).**

soils and minimum under paddy-oilseed soils, following a trend; forest >pasture >apple >saffron >paddy-oilseed. Further, a reduction in POC was recorded along the soil depth and all depths differed significantly ($p < 0.05$). The elevated POC levels in forest soils can be linked to larger litter deposits, which contain additional labile carbon (*Laik et al., 2009*; *Barreto et al., 2011*) thus, promote microbial vitality and quantity. In addition, it could also be linked to negligible anthropogenic disturbances, surface cover, and mechanical soil maintenance to decrease erosion. Comparable findings were documented in the central Himalayan range by *Kalambukattu et al. (2013)*, who found that undisturbed land use categories exhibited greater POC owing to carbon buildup that is preserved by soil aggregates. The lowest POC under cultivated soils might be attributed to the deterioration of soil aggregates through tillage that exposes POC to increased breakdown and mineralization

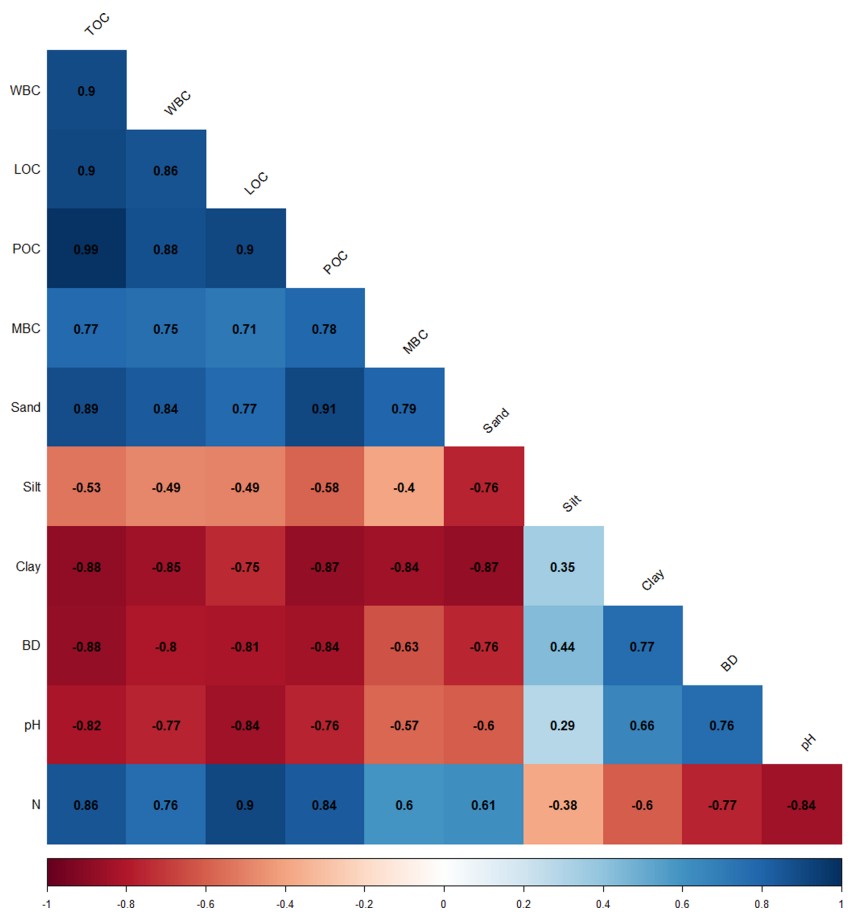

**Figure 4** Correlation between physico-chemical characteristics and SOC pools.

resulting in decreased POC levels (*Bayer et al., 2006*). Furthermore, this fraction does not establish organo-complexes with minerals, thereby making it prone to mineralization (*Christensen, 1988*). These observations are further in agreement with *Six et al. (1998)*, *Figueiredo, Resck & Carneiro (2010)*, *Liu et al. (2014)*, *Sofi et al. (2016)*, and *Poeplau & Don (2013)*.

The mean values of MBC under surface (0–30 cm) soils ranged from 830.21–1,458.12 mg kg$^{-1}$ and in sub-surface (30–60 and 60–90 cm) layers from 546.38–1,232.53 and 365.50–1,004.14 mg kg$^{-1}$, respectively across studied LUSs (Table 6). The amount of MBC differed significantly ($p < 0.05$) among studied LUSs with maximum content under forest soils and minimum under paddy-oilseed soils. MBC content decreased significantly with soil depth and all depths differed significantly ($p < 0.05$). The size of the microbial reservoir is influenced by LUSs and the management activities of soil. In addition to the above-mentioned reasons, the elevated MBC values recorded under forests could be ascribed to the synthesis of biomass in the rhizosphere and to a lesser extent, reduced soil tillage. The biomass activity of soil might have been enhanced by organic matter inclusion and improved nutrient cycling in forests (*Min et al., 2003*). The lowest MBC

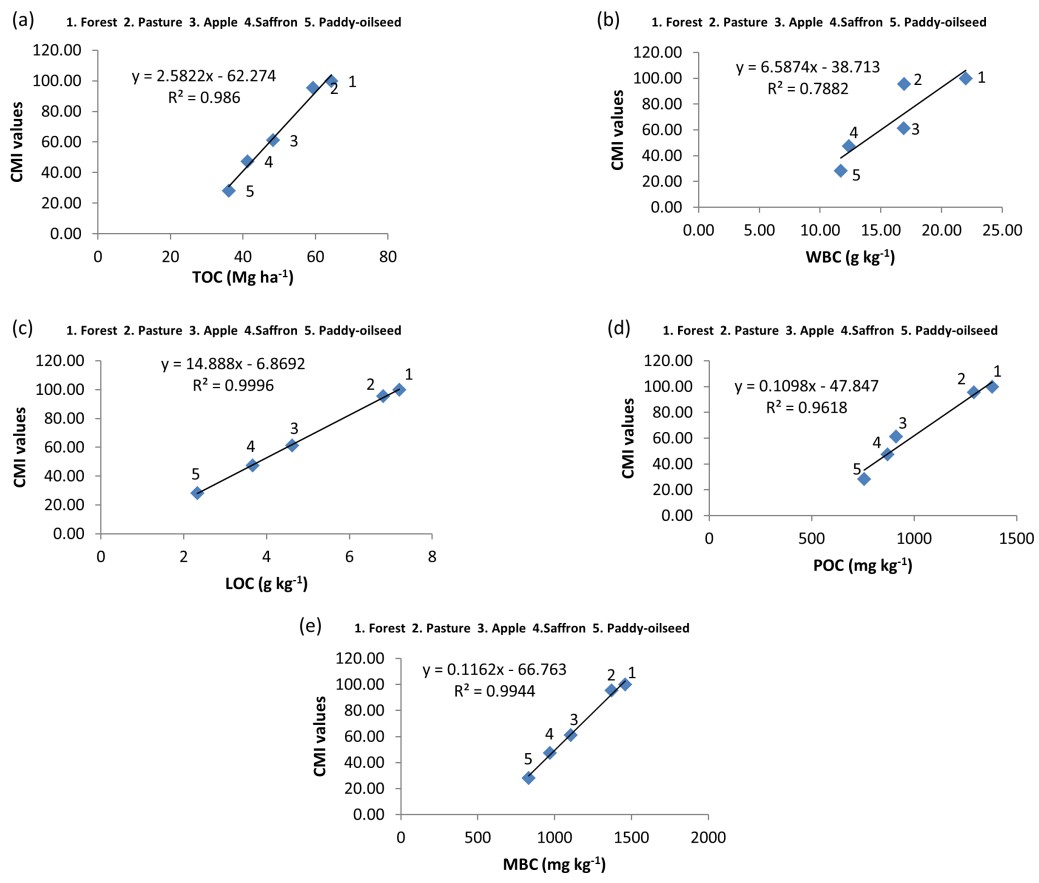

**Figure 5** Relationship among SOC pools and CMI at 0–30 cm depth.

under cultivated soils might be attributed to low vegetation, tillage operations, oxidation of OM and microbial OC declines steadily with a distance from the rhizosphere (*Paul & Clark, 1996*). These observations are concordant with those of *Beare et al. (1994)*, *Purakayastha, Smith & Huggins (2009)*, *Huang & Song (2010)*, *Singh et al. (2011)*, and *Sofi et al. (2016)*.

## Carbon management index (CMI)

The evaluation of CMI was performed for studied LUSs while taking forest surface soil as a reference for calculation. A significant influence of LUSs and soil depth on CMI values was observed. In this study, the highest CMI values were observed under forest followed by pasture and lowest in paddy-oilseed with the following trend: forest(100) >pasture (95.46) >apple (61.16) >saffron (47.44) >paddy-oilseed (28.29) at surface soils, whereas, CMI showed a significant decline in lower depths across studied land use systems (Fig. 3). This reflects that CMI serves as reliable metrics for estimating variations in SOC pools as well as soil quality. Forest soils had the highest CMI value, which was different from other land uses (Fig. 3), since forest provides a less oxidative environment for the breakdown of POC and POC build-up has been enabled by the prevalence of the thicket awning, a preventive pattern of the macro-aggregates and lesser erodability. These observations are similar to

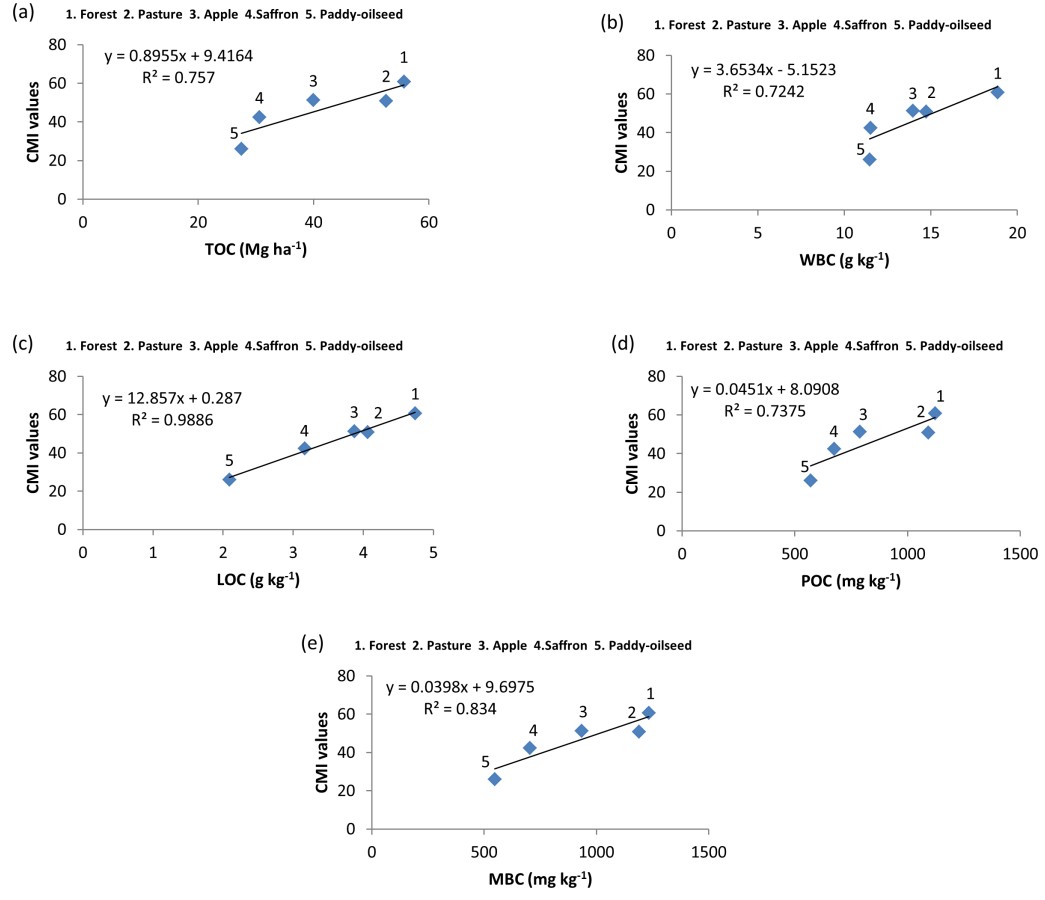

**Figure 6 Relationship among SOC pools and CMI at 30–60 cm depth.**

those noticed by *Blair, Lefroy & Lisle (1995)*. The periodic organic matter preservation and reduced depletion concerning forest and pasture have proven to augment the values of CMI (*Blair & Crocker, 2000*). The lower level of CMI in paddy-oilseed soils indicates that this land use type has lower inputs of carbon and raised turnover levels because of increased temperature and SOC depletion. Similar reports have been noticed by *Cao et al. (2013)*. The usage of nitrogen-based fertilizer has improved biomass thereby aggrandizing SOM. These observations are similar to *Vieira et al. (2007)* who noticed fertilization and residue inclusion aggrandize the SOM's lability by 12–46%, thus raising CMI. There exists no definite norm for CMI since it relies on the indigenous LUS of the region; therefore, greater values of CMI signify carbon restoration whereas the least CMI levels denote the depletion of carbon. This is further consistent with reports of *Benbi et al. (2015)*.

## Relationship matrix of physico-chemical characteristics, SOC pools and CMI

A correlation was performed between the physico-chemical characteristics and SOC pools which revealed a positive significant association of N with carbon pools, whereas pH and

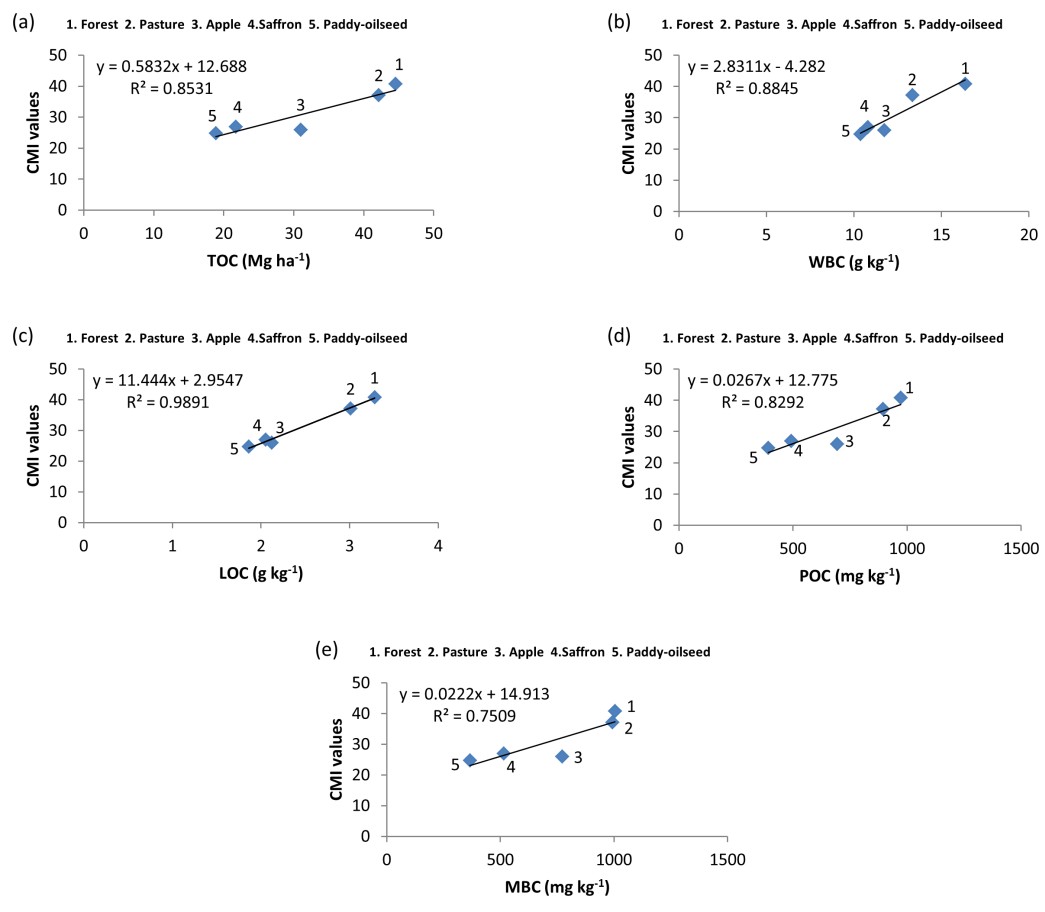

**Figure 7 Relationship among SOC pools and CMI at 60–90 cm depth.**

BD showed a negative significant association with SOC pools (Fig. 4). From regression analysis (Figs. 5, 6 and 7), a positive significant association (high R-squared values) was marked between CMI and different organic carbon pools in surface (0–30 cm) as well as sub-surface (30–60 and 60–90 cm) layers.

## CONCLUSION

The study concluded that various LUSs and soil depth had an impact on SOC pools and, eventually, the CMI. Soil carbon pool dynamics were better in the case of forest and pasture soils while agricultural perturbations including stubble elimination, tillage, and increased erosion could be a substantial driver of carbon depletion from cultivated soils. However, among cultivated land uses, apple soils showed a better carbon management index. Therefore, land use conversions must be checked to impede carbon depletion. A decrease in organic carbon fractions with change in management practices and soil depth has been observed. Minimizing erosion, the addition of more organic matter, diversifying cropping systems, and minimum or zero tillage operations are important steps in reversing soil depletion and improving soil quality. Therefore, we must adopt best management

practices to sequester more carbon which eventually improves the quality of soil and ecological balance and ultimately serve the purpose of global sustainability.

### Funding
The authors received no funding for this work.

### Competing Interests
The authors declare there are no competing interests.

### Author Contributions
- Yasir Hanif Mir conceived and designed the experiments, performed the experiments, prepared figures and/or tables, authored or reviewed drafts of the article, and approved the final draft.
- Mumtaz Ahmad Ganie conceived and designed the experiments, performed the experiments, prepared figures and/or tables, and approved the final draft.
- Tajamul Islam Shah conceived and designed the experiments, prepared figures and/or tables, and approved the final draft.
- Aziz Mujtaba Aezum conceived and designed the experiments, performed the experiments, prepared figures and/or tables, authored or reviewed drafts of the article, and approved the final draft.
- Shabir Ahmed Bangroo conceived and designed the experiments, prepared figures and/or tables, authored or reviewed drafts of the article, and approved the final draft.
- Shakeel Ahmad Mir analyzed the data, prepared figures and/or tables, and approved the final draft.
- Shahnawaz Rasool Dar conceived and designed the experiments, prepared figures and/or tables, and approved the final draft.
- Syed Sheeraz Mahdi analyzed the data, prepared figures and/or tables, and approved the final draft.
- Zahoor Ahmad Baba analyzed the data, prepared figures and/or tables, and approved the final draft.
- Aanisa Manzoor Shah analyzed the data, prepared figures and/or tables, and approved the final draft.
- Uzma Majeed analyzed the data, prepared figures and/or tables, and approved the final draft.
- Tatiana Minkina analyzed the data, prepared figures and/or tables, authored or reviewed drafts of the article, and approved the final draft.
- Vishnu D. Rajput analyzed the data, prepared figures and/or tables, and approved the final draft.
- Aijaz Ahmad Dar analyzed the data, prepared figures and/or tables, and approved the final draft.

## Data Availability

The raw data is available in the Supplemental File.

## Supplemental Information

Supplemental information for this article can be found online at http://dx.doi.org/10.7717/peerj.15266#supplemental-information.

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
