# Peer review of "Soil organic carbon pools and carbon management index under different land use systems in North western Himalayas"

_PeerJ, doi:10.7717/peerj.15266_

## Round 0.1 · original submission · Major Revisions

Major revisions are required. Reply to both reviewers' comments, the revised paper will be shared with the reviewers again, thanks.

·

Basic reporting

This manuscript is clearly written and well organized. The introduction and background are reasonable and sufficient. The figures and tables are of good quality and clear. Raw data was provided for the majority of measurements. But there’s no data on the Lability of Carbon in raw data. I suggested the author add LC to the raw dataset.

Experimental design

In general, the experimental design was reasonable and clearly presented. Additions and modifications would be suggested as follows:
1) In section 2.3. Carbon management index, no detail on how the C remaining un-oxidized by KMnO4 was estimated. Was it measured or estimated from the difference between the total organic C pool and the labile C? I suggested the author add the details on LC calculation and provide the data in the raw dataset.
2) It is not clear how the data replication is achieved. Three soil samples were collected at each sampling location or from different locations for each land use type? Or it’s just one sample tested three times? I suggest explicitly indicating how the replication was achieved, and also reporting the standard error of the mean for each parameter.
3) Basic physico-chemical parameters of soil under different land uses were not reported. Better to present the physiclo-chemical parameters of soils.

Validity of the findings

The results are reasonable given the experiments and data. Some minor modifications were needed.
1) The data reported in Table 1 for TOC (in unit of Mg ha-1) is TOC stock which is the TOC mass per area for a given depth. Only TOC concentration measurement is described in the method. I suggest authors add an equation for TOC density calculation.
2) It’s not clear whether two-way or two-way with interaction ANOVA analysis was used for statistical analysis. There is an interaction term for C.D in tables 1-5, but I checked AIC for the TOC model, and it seems the two-way without interaction model is the best fit. It would be good to indicate whether the interaction of land use and depth is included in ANOVA analysis and how the C.D was calculated to help verify the results.

Additional comments

Suggested correcting a few linguistic errors. See examples as follows:
1) Line 81 “comprises net sown area…”,
2) Line 55 “In addition, conservation tillage and agro-forestry has been …”

Reviewer 2 ·

Basic reporting

Comments to “Soil organic carbon pools and carbon management index under different land use systems in North Western Himalayas”

In the present manuscript, the authors report results concerning the influence of land use and depth on soil organic carbon (SOC) pools in NW Himalayas. In detail, 6 land uses and 3 depths were investigated, as well as different SOC “forms” analysed.
The paper is generally clear and well written. The topic is not new, but still of great interest.

Experimental design

Unfortunately, I do not consider this work as original enough and of added value that deserves publication in this journal, at least in the present status. Tons of papers have been published on the same topic across different regions of the world (e.g., Eur J Soil Sci 45, 449–458, 1994; Glob Chang Biol 8, 345–360, 2002; Glob Chang Biol 18, 2233–2245, 2012; Geoderma 192, 189–201, 2013; Biol Fert Soils 54, 671–681, 2018; Geoderma 353, 423–434, 2019); unfortunately, the Authors did not underline sufficient innovative aspects that this work provides with respect to what is already present in scientific literature.
Moreover,
1) the paper is not well organized, balanced and to the point. The Introduction lacks a real state-of-the-art on the topic, while the discussion is very long and included paragraphs that should be moved to the Introduction. The M&M is extremely short and analytical details are completely missing. Finally, although the authors reported 77 references, fundamental papers on the topic are not listed, while some articles on exotic journals and proceedings are cited.
2) I also think that results are not always supported by data, as I have several concerns about both the experimental approach and their interpretation. In detail,
a) it is not clear to me the difference between TOC and WBC, being the Walkley-Black a method to determine TOC. I had a look to the paper cited (i.e., Snyder and Trofymow, 1984), but the authors used a different method to determine the same parameter. Both are TOC, but according to different approaches. Therefore, why results are different?
b) the authors investigate 6 land uses, but without replicates. My question is: Is each single land uses really representative? I mean, it would be much better if a lower number of land use systems were analysed but e.g., in triplicates.
c) soil samples were not characterized at all… Which is the role of texture, mineralogy, N content? All these parameters have a big inpact on SOC accrual and distribution.
d) obtained data about SOC “pools” (or forms) are somehow interesting, but they should not be discussed separately…

Validity of the findings

In few words, I am very sorry but the submitted paper needs quite major revisions before being accepted. I would suggest rejection with the option of resubmission.

---

## Round 0.2 · Minor Revisions

Please address the minor revisions noted by the reviewers.

·

Basic reporting

This manuscript is clearly written and well organized. The introduction and background are reasonable and sufficient. Minor revision is suggested as follows:
1). I suggest authors provide high-resolution graphs for Figures 1-7.
2). The figures titles for Figures 1-7 should be at the bottom of the figures instead of at the top of them.

Experimental design

All the comments on the experimental design have been well addressed. The experimental design was reasonable and clearly presented.

Validity of the findings

Most comments have been well addressed. The following minor revision is still suggested before acceptance.
1). The authors claimed that Two-wany ANOVA with interaction analysis was used for statistical analysis. Was goodness of fit estimated to select the best-fit model (One-way, Two-way, or Two-way with interaction)? Please provide information that supports the selection.
2). How was the significance of the difference determined in Table 2-Table 6? The letters were labeled at the “Factor Mean”, so I guess the statistical analysis was based on the “Factor Mean”? How is the significant difference result if the analysis is based on each value for LUS and depth? (which is typically used in the statistical analysis instead of “factor mean”)

Reviewer 2 ·

Basic reporting

The authors improved the submitted paper, addressing most of the comments provided by the reviewer.

l. 24: delete the first “soil”
l. 24: replace “Walkley and black” with “Walkley-Black”
l. 28: “…depth of 90 cm”
l. 34-35: replace “soil organic carbon” with “SOC”. Check throughout the paper..
l. 45: “Kooch et al. 2020”. This ref is not useful for this statement and can be deleted
l. 61: replace “soil’s” with “soil”
l. 64-65: The original paper should be cited for this statement, not “Chen et al. 2019; Yadav et al. 2019”. See Lal (2001). Climatic Change, 15, 35-72. https://doi.org/10.1023/A:1017529816140
l. 71: “Vezzani 2001” can be deleted,… as not accessible to the reader
l. 75: a reference is needed for this statement. The following is suggested “J Environ Monit 14, 2438-2446, 2012”
l. 320: delete “References”

7 figures and 6 tables are really too much… it would be nice if few of them are condensed.

Experimental design

no comment

Validity of the findings

no comment

Additional comments

no comment

---

## Round 0.3 · accepted · Accept

The manuscript is improved and accepted by both reviewers.

·

Basic reporting

This manuscript is clearly written and well organized. The introduction and background are reasonable and sufficient. The figures and tables are of good quality and clear. Raw data was provided for the majority of measurements.

The titles of the Figures 1-7 were still on the top of the figures instead of at the bottom though the authors claimed that the modifications were made. Need to address in the final format.

Experimental design

No further comments.

Validity of the findings

No further comments

Additional comments

No further comments

Reviewer 2 ·

Basic reporting

The authors improved the submitted paper, addressing all minor comments provided by the reviewer.

Experimental design

no comment

Validity of the findings

The authors improved the submitted paper, addressing all minor comments provided by the reviewer.